# Effectiveness–implementation hybrid type 2 study evaluating an intervention to support 'information work' in dementia care: an implementation study protocol

Chiara De Poli ,[1] Jan R Oyebode,[2] Christopher Binns,[3] Richard Glover,[4] Mara Airoldi[5]

[1]Department of Social Policy and Department of Management, London School of Economics and Political Science, London, UK
[2]School of Dementia Studies, University of Bradford, Bradford, UK
[3]Tees, Esk and Wear Valleys NHS Foundation Trust, Durham, UK
[4]NHS North of England Commissioning Support Unit, Durham, UK
[5]Blavatnik School of Government, University of Oxford, Oxford, UK

**Correspondence to**
Chiara De Poli;
c.de-poli@lse.ac.uk

## ABSTRACT

**Introduction** Patients with long-term conditions consistently report a lack of information around services and support available to them. This unmet need for information is significant among people with dementia and family carers. A quality improvement intervention is being carried out to tackle this issue as part of a co-creation initiative in the North East of England (UK). The intervention consists of the dissemination (via the local Community Mental Health Services for Older People) of a leaflet about services available to people with dementia and their family carers in the study site. This protocol is reported in accordance with the Standards for Reporting Implementation Studies.

**Methods and analysis** This effectiveness–implementation hybrid type 2 study aims at understanding (1) the unfolding and outcomes of the implementation strategy, (2) the outcomes of the intervention (for people with dementia and family carers, staff implementing the intervention and local service providers) and (3) the contribution of co-creation to the design and implementation of the intervention and its outcomes. The prospective theory of change of the intervention articulated by local stakeholders is used as a reference framework against which to assess the implementation and outcomes of the intervention. Evaluation data will be collected through in-depth interviews with people with dementia and family carers receiving the intervention, staff implementing the intervention and managers from local service providers. Referral data from local service providers will be collected to triangulate the interview data. A focus group with key stakeholders will support the sense-making of findings. The realist configuration of mechanism–context–outcome, operationalised using an information behaviour model, will inform data analysis and interpretation.

**Ethics and dissemination** Ethical and research governance approvals have been obtained from the West Midlands—South Birmingham Research Ethics Committee. The results of the study will be submitted for publication in peer-reviewed journals and disseminated through conferences.

## Strengths and limitations of this study

► The study is reported in accordance with the Standards for Reporting Implementation Studies framework complemented by additional relevant reporting standards, such as the Template for Intervention Description and Replication checklist and Proctor *et al*'s framework for specifying and reporting implementation strategies.
► The evaluation framework integrates the evaluation of the implementation strategy and of the implemented intervention both in terms of process and of outcomes.
► The mixed-method evaluation uses data collected through in-depth interviews with three sets of stakeholders, services' referral data and a focus group.
► In-depth, longitudinal interviews with people with dementia are planned to explore their experiences of receiving and using the leaflet; however, recruitment can be difficult and dropout rate can be high.
► The evaluation will provide insights regarding the process and impact of delivering and receiving the intervention but will have no comparison condition.

## INTRODUCTION

Patient activation, self-management, ageing in place, enablement and shared decision-making are recurrent themes in policy discourses across health systems addressing ageing populations, chronic diseases, multimorbidity and complex needs.

Underpinning, although often implicit, assumptions of such policy discourses is the idea that by providing information, patients and family carers are supported to make better choices, have more agency on managing their own health (or the health of the person they care for), may feel empowered and ultimately may report better outcomes and satisfaction with the care and support they receive.

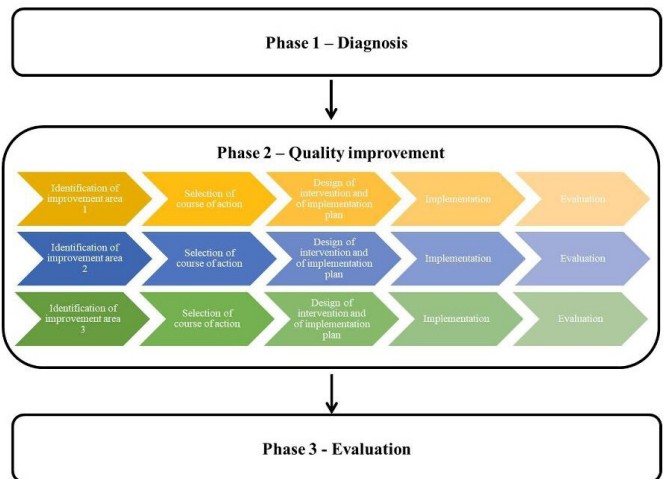

**Figure 1** The phases of the co-creation initiative.

However, such a view fails to acknowledge the 'information work', that is, the process of seeking, using, evaluating and sharing of information[1] that is required from patients and family carers in order for them to be able to make better choices or to play a more active role in their own or others' care.[2] Also, it assumes that patients and family carers are physically and cognitively able to carry out information work, and it overlooks the emotional processes that influence adaptation to long-term conditions, which may influence whether information work is successfully undertaken.[3]

Extant research both documents and describes the unmet information need of patients and family carers and also offers possible explanations for it.[4–6] This issue has been recognised as highly significant among people with dementia and family carers,[7–11] whose information needs span beyond the topics usually addressed by information materials available, leaving them with some information gaps.[12 13] Moreover, people living with dementia identify a preference to receive information directly from health and social care professionals rather than over the internet, which is the primary source of information to which they are often signposted.[14] They highlight that the timeliness of their information needs, which vary along the dementia trajectory,[15] is seldom taken into account by information providers and resources. They also expect up-to-date information to be provided in a culturally acceptable way and in easy to understand language.[9 16 17] Several user-centred models in the field of information science have been developed to unpack users' information behaviour and explore how the actual process of information need, seeking and use occurs, in order to explain the observed variations in information need.[18] Notwithstanding the specificities of each, these models consistently acknowledge that information is context sensitive and uniquely dependent on the individual user, who actively constructs its subjective meaning.[19]

Building on this research, an implementation study is being carried out as part of a co-creation initiative in dementia care taking place in the North East of England

(UK). The study aims at designing, implementing and evaluating a leaflet about local services providing care and support to people with dementia and their family carers living in the study site.

This article reports the protocol for the evaluation of the implementation study in accordance with the Standards for Reporting Implementation Studies (StaRI) framework and adopts the recommended dual strands for reporting, one about the intervention being implemented and the other about its implementation strategy (ie, the 'bundle' of techniques used to enhance the adoption, integration into routine practice and sustainability of the intervention).[20 21]

The evaluation has three aims. First, it intends to review the unfolding and outcomes of the implementation strategy. Second, it aims to assess the outcomes of the intervention as perceived by people living with dementia and their family carers, staff implementing the intervention and local organisations providing care and support to people with dementia and family carers and to explain how and why the intervention contributed to achieve these outcomes. Third, it will reflect on the extent to which, how and why the co-creation initiative has influenced the design, implementation and impact of the intervention.

## INTERVENTION DEVELOPMENT

The implementation study herein described has been developed as part of a co-creation initiative in dementia care taking place in the North East of England (UK). The initiative, started in 2015 and with expected completion at the end of 2020, involves a constellation of local stakeholders, including people living with dementia and their family carers, health and care professionals, service managers and commissioners alongside a research team, encompassing university-based researchers and a local research facilitator.

Grounded in the principles of action research, co-creation activities were organised in three broad phases (figure 1).[22] The diagnostic phase (phase 1, 2015–2018) mapped the local dementia care system and identified its strengths and weaknesses. The improvement phase (phase 2, 2017 and 2018) articulated and designed three interventions to address the priority issues identified in the diagnostic phase. Each improvement area was addressed via a subcycle of action research. Local stakeholders with an investment in the respective improvement areas were invited to work with the research team to identify different available options to address each area, select a course of action and accordingly design an intervention and define its implementation.

The planned evaluation (phase 3, 2020) will focus on both the improvement interventions and the co-creation initiative. With respect to the first, the evaluation aims to assess the outcomes of the implementation strategy and the outcomes of each intervention. With respect to the latter, the evaluation aims to unpack how the co-creation

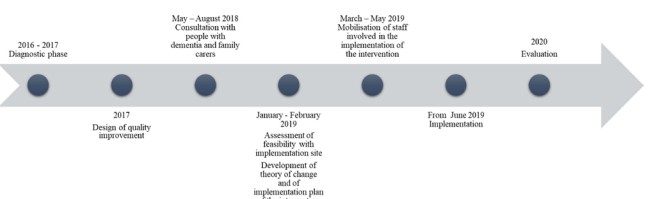

**Figure 2** Timeline of the development of the intervention.

initiative has unfolded in practice and its impact on local policy-making around care and support for people with dementia.

### The intervention and the implementation strategy

A fundamental unmet need for information was reported by people with dementia and family carers who took part in the diagnostic phase. Local stakeholders agreed to tackle this issue by providing up-to-date, accessible and timely information about local services available to those living with dementia and family carers. Figure 2 provides a timeline of the process underpinning the development of the agreed intervention.

A local working group (including commissioners, statutory service providers, third sector organisations providing support and care to people with dementia and family carers, a professional designer, alongside the research team) was set up with the remit of developing an information leaflet and planning its implementation. Efforts to involve family carers and people with dementia by asking local third sector organisations to identify people keen to be involved in the working group resulted in one family carer joining the group. Another turned down the offer because of competing commitments and no one living with dementia was identified. To mitigate their absence, inputs from front-line staff providing daily direct support and care to people with dementia and to family carers were used (one was also a family carer). In a similar vein, we relied also on the professional designer who had extensive experience and expertise of designing information material intended to be used by people with cognitive impairment.

The group produced three prototypes which were shared and discussed in a consultation exercise involving about 20 among people with dementia and family carers. The consultation exercise was planned around the schedule of activities or groups organised locally for people with dementia and/or family carers. This solution overcame some of the practical problems that might have limited their participation in the working group.

The preferred prototype was iteratively refined using best practice guidance on Patient Information Leaflets[23] and taking into account feedback provided by its potential users, commissioners and service providers. Following local discussion, it was agreed to design only one version of the leaflet for both family caregivers and individuals living with dementia. The leaflet was intended to cover essential information about, and contact details of, the

key local organisations that provide dementia-related support and services, which, in practice, often cater for both groups. The intention was to signpost people to such organisations, rather than providing an exhaustive list of services offered. Hence, two versions of the leaflet would not have added substantially different content but could have contributed to information overload, sometimes experienced by those living with dementia and their families (eg, around the time of receiving the diagnosis). In terms of graphics, the design aimed to maximise accessibility, irrespective of the intended user's group. From the implementation perspective, it was felt that it would be easier to embed one version of the leaflet in the practice of professionals and, equally, designing and using one version of the leaflet could, more likely, contribute to its sustainability in the longer term, for example, by simplifying the updating process.

In its final version, the leaflet presents services clustered around four main themes ('I want to speak to someone for advice and support', 'I want to live safely and with the right support', 'I want to know what to do in a crisis' and 'Where can I go for more information?') and it can be printed out on a two-sided A4 using a high contrast colour combination.

Subsequently, the working group agreed a phased implementation process: the Community Mental Health Services for Older People (CMHSOP) within the local mental health trust were identified as the first implementation route because of their key role in diagnosing dementia and in the resulting postdiagnostic care planning. Following diagnosis and initial care planning, those whose needs are stable are discharged to primary care, while those with more complex needs continue to receive secondary services.

The working group preliminarily explored with the CMHSOP service manager, team managers and representatives of the community psychiatric nurses (CPNs) how the leaflet could be used in clinical practice. In January 2019, following positive and supportive feedback, the leaflet working group members and the CMHSOP service manager, team managers and two CPNs took part in a focus group aimed at articulating the theory of change associated with the intervention itself. This exercise helped making explicit the expected process of implementing the intervention in practice and its anticipated impact. Mobilisation activities (eg, information sessions within team meetings and regular reminders emailed to team managers and cascaded to CPNs) were planned and carried out to inform, and subsequently to remind, all CPNs about the leaflet and its use in practice. In June 2019, all the CPNs across the four locality teams covering the study site were invited to start using the leaflet in their clinical practice.

CPNs agreed to hand out the leaflet to patients on their caseload who received a diagnosis of any type of dementia and/or to their family carers. CPNs were expected to use their clinical judgement in deciding to whom and when

to offer the leaflet, dependent on the patient's readiness to receive the leaflet and their individual and family circumstances. They expected to hand it out during their encounter with patients and families, when they felt that the leaflet would support their conversations with them. For these reasons, the use of the leaflet was not embedded at a specific point of the clinical pathway.

A detailed description of the intervention and its implementation strategy are outlined in accordance with the StaRI framework, respectively in tables 1 and 2 (item 9).

## METHODS AND ANALYSIS
### Evaluation framework

The study adopts an effectiveness–implementation hybrid type 2 design, which allows simultaneous evaluation of the intervention and its implementation strategy,[24] as reflected in the evaluation framework (figure 3).

The evaluation framework maps the phases of the study (design–implementation–evaluation) on to the key components of the intervention and shows the relations among them. In the design phase, the intervention and the implementation strategy (ie, the bundle of professional and material interventions intended to sustain the adoption and use of the leaflet among CPNs) were articulated with the theory of change focus group. Following the implementation, the evaluation will:

1. Assess the unfolding of the implementation strategy and its outcomes.
2. Assess the outcomes of the intervention as perceived by people with dementia and family carers, the implementers of the intervention, as well as the providers of local services available to people with dementia and their families.
3. Identify the mechanisms of action, that is, the 'active ingredients' that ignited the implementation strategy and the intervention and, hence, contributed to the observed outcomes.
4. Reflect on the contribution of context (eg, the social, economic, policy and organisational factors surrounding the intervention, including, but not limited to, the co-creation initiative, as described in table 1, item 7) to the design and implementation of the intervention and its implementation strategy and relative outcomes.

### Outcomes of the implementation strategy

The implementation strategy is evaluated with respect to both its outcomes and its process dimension, as per the StaRI framework. Following Proctor *et al*'s taxonomy,[25] the following outcomes will be assessed with respect to the implementation strategy (table 1, item 11):

► Acceptability of the intervention by CPNs: perception among CPNs that the intervention is agreeable, palatable or satisfactory.
► Adoption of the intervention by CPNs: CPNs' intention, initial decision or action of trying out the intervention.

► Appropriateness of the intervention implemented as judged by CPNs: perceived fit, relevance or compatibility of the intervention for the setting of the CMHSOP and perceived fit of the intervention to address the issue.
► Cost of the intervention for the implementers.
► Feasibility of the implementation of the intervention: the extent to which the intervention can be successfully used or carried out within the setting of the CMHSOP.
► Penetration of the intervention: the extent to which the intervention is integrated into the practice of the CMHSOP.
► Sustainability of the intervention: extent to which the intervention is maintained or embedded within the CMHSOP.

### Outcomes of the intervention

The theory of change exercise run in January 2019 highlighted that the intervention was expected to have an impact on (table 2, item 11):

► Its main target population, that is, people with dementia and family carers (direct outcomes).
► The implementers of the intervention, that is, CPNs, team and service managers of the CMHSOP delivering the intervention (proximal outcomes).
► Local providers of care and support to people with dementia and their families, such as local third sector organisations (distal outcomes).

With respect to people with dementia and family carers, the intervention was expected to improve three outcomes directly associated with the intervention: (1) their satisfaction with the availability of information on local services (eg, in terms of timing and quality of information), (2) their awareness of local services and (3) in the longer term, their access and use of local services.

With respect to the implementers, the intervention was expected to influence two proximal outcomes : (1) CPN, team and service manager's awareness of local services and support available to people diagnosed with dementia and to their family carers and (2) CPN, team and service manager's confidence to hold conversations with people living with dementia and their family carers about seeking support and using local services.

With respect to local service providers, the intervention was expected to impact on two distal outcomes: (1) volume of referrals into their services and (2) appropriateness of the use of services among those living with dementia and family carers (eg, with respect to the type and level of need).

### Process evaluation

The Medical Research Council (MRC) framework for process evaluation[26] is used to investigate (tables 1 and 2, item 12):

► The actual implementation strategy, in terms of implementation fidelity (or, conversely, adaptation), reach and dose.

**Table 1** Overview of the implementation strategy reported in accordance with the Standards for Reporting Implementation Studies checklist

**Implementation strategy (ie, how the intervention was implemented)**

| Item | | Description and reference to methodological frameworks and reporting standards |
|---|---|---|
| 1. Title | | Effectiveness–implementation hybrid type 2 study evaluating an intervention to support 'information work' in dementia care |
| 2. Abstract | | See Abstract of the protocol |
| 3. Introduction | To identify the problem | See Introduction of the protocol |
| 4. Introduction | Rationale for the implementation strategy and the intervention | See Intervention development of the protocol |
| 5. Aims and objectives | | 1. To assess the unfolding of the implementation strategy and its outcomes |
| | | 2. To reflect on the extent to which, how and why the co-creation initiative has influenced the design of the implementation strategy and its unfolding and outcomes |
| 6. The design and key features of the evaluation | | The evaluation of the implementation strategy aims at understanding the feasibility of the intervention by investigating how the actual implementation process has unfolded in practice and identifying any discrepancies between the planned and the actual implementation process |
| 7. Context | Context in which the intervention is implemented | Community Mental Health Services for Older People (CMHSOP) within a mental health foundation trust in the North East of England (UK) ▲ North East of England is an under-researched area compared with other parts of England ▲ Local health economy with average recorded diagnosed dementia prevalence (for 2018 and 2019, local recorded prevalence rate 0.9% vs national recorded prevalence rate 0.8%) and average recorded annual review rate (for 2018 and 2019, local annual review rate 79.2% vs national recorded prevalence rate 77.9%)[42] ▲ Mental health foundation trust, established in 2006 providing a range of mental health, learning disability and eating disorder services for around 2 million people across a large and diverse geographical area ▲ CMHSOP providing community-based services for older people experiencing cognitive impairment and mental health problems. Organised around four localities, each locality team includes community psychiatric nurses (CPNs), occupational therapists, physiotherapists, psychologists, support workers to provide multidisciplinary support ▲ Intervention was developed in the context of a 5-year action research co-creation initiative funded by an English charity, involving researchers from three English universities and a research facilitator from the local commissioning support unit alongside a constellation of local stakeholders |
| 8. Target sites | | Four locality teams, staffing levels and caseload are reported in the table below |

| Team | Total staff | CPNs | Caseload |
|---|---|---|---|
| Team 1 | 48 | 9 | 1368 |
| Team 2 | 43 | 8 | 1377 |
| Team 3 | 27 | 5 | 787 |
| Team 4 | 26 | 5 | 791 |

Continued

**Table 1** Continued

## Implementation strategy (ie, how the intervention was implemented)

| Item | Description and reference to methodological frameworks and reporting standards | |
|---|---|---|
| 9. Implementation strategy | Description of the implementation strategy in accordance with Proctor et al's framework for reporting implementation strategies[43] | **Name and definition of the implementation strategy:** bundle of professional and material interventions aiming at the development, adoption and use of the dementia leaflet among CPNs (based on framework by Mazza et al and Powell et al[44 45]

**Specification of the implementation strategy**

▲ Actors: CPNs, team managers and service manager across four CMHSOP teams
▲ Actions: Described in accordance with relevant phase of the Plan-Do-Study-Act framework
▲ Plan:
 – Professional interventions
 – Established a local working group in charge of overseeing the development, the leaflet and its implementation strategy
 – Iteratively developed a dementia leaflet, embedding feedback from people with dementia, family carers, professionals from local organisations providing care and support to people with dementia and family carers
 – Identified the implementation strategy via CPNs based in CMHSOP, identified local champion (service manager), identified implementation sites
 – Organised mobilisation activities with implementation sites, including information sessions at team meetings and email reminders
 ▲ Material intervention
 – Printed copies of the leaflet made available to implementers
▲ Do: CPNs offer a copy of the leaflet to people with dementia or their family carers attending an appointment (any type)
▲ Action targets: People living with dementia (any type of dementia, diagnosed at anytime) and their family carers attending an appointment with their CPNs. CPNs were expected to use their clinical judgement in identifying patients or family members deemed to be ready to receive the leaflet
▲ Temporality: The leaflet can be offered at any point during the period in which the patient was under the care of the CMHSOP (at initial assessment, at post diagnosis, at reviews and at discharge)
▲ Dose: Depending on the individual readiness to receive the leaflet and individual and family circumstance, CPNs decide whether to offer or not the leaflet as part of their appointment with the patient. The leaflet could be offered more than once at different points in time |
| 10. Subgroups | Any subgroups recruited for additional research tasks, and/or nested studies | Not applicable to this study |
| **Evaluation** | | |
| 11. Outcomes | Using Proctor et al's taxonomy of outcomes for implementation research[25] | ▲ Acceptability of the intervention by CPNs: perception among CPNs that the intervention is agreeable, palatable or satisfactory
▲ Adoption of the intervention by CPNs: CPNs' intention, initial decision or action to try the intervention
▲ Appropriateness of the intervention implemented as judged by CPNs: perceived fit, relevance or compatibility of the intervention for the setting of CMHSOP and perceived fit of the intervention to address the issue
▲ Cost of the intervention for the implementers
▲ Feasibility of the implementation of the intervention: the extent to which the intervention can be successfully used or carried out within the setting of the CMHSOP
▲ Penetration of the intervention: integration of the intervention within the CMHSOP
▲ Sustainability of the intervention: extent to which the intervention is maintained or institutionalised within the CMHSOP |
| 12. Process evaluation objectives and outcomes related to the mechanism(s) through which the strategy is expected to work | Objectives and outcomes related to the mechanism by which the strategy is expected to work, using the Medical Research Council (MRC) framework for complex interventions[26] | ▲ Fidelity of the intervention: adherence to the intervention protocol
▲ Adaptations of the intervention
▲ Dose: the quantity of intervention implemented
▲ Reach: Whether the intended audience comes into contact with the intervention, and how |
| 13. Data collection methods | | In-depth interviews with CPNs, team managers and service manager of CMHSOP |

Continued

| Table 1 | Continued | | | |
| --- | --- | --- | --- | --- |
| **Implementation strategy (ie, how the intervention was implemented)** | | | | |
| Item | Description and reference to methodological frameworks and reporting standards | | | |
| 14. Sample size | ▲ Up to 12 CPNs from 4 locality teams<br>▲ Four team managers<br>▲ One service manager | | | |
| 15. Methods of analysis | Thematic content analysis of interview data | | | |

► The mechanisms of action of the implementation strategy and of the intervention which brought about change.

The process evaluation will highlight how the implementation process has unfolded in practice and any discrepancies between the actual implementation process and how it was articulated in the theory of change. Moreover, it will help to surface how and why the intervention and its implementation strategy have contributed to achieve these outcomes.

### Analytical framework

The framework underpinning the realist evaluation will be adopted as analytical tool. Realist evaluation acknowledges that interventions work (have successful outcomes) in so far as they introduce appropriate ideas and opportunities (mechanisms) to groups of stakeholders in the appropriate social and cultural conditions (context), as condensed in the following formula[27] mechanism+context=outcome.

According to the realist framework, mechanisms are the underlying processes or hidden causal levers of intervention activities that make them work[28] and are represented by the psychological or social explanations of behaviour of intervention participants and their ideas and reasoning about how change is achieved.[29 30] Context refers to the conditions likely to enable or constrain the activation of interventions' mechanisms (eg, the organisational setting or the broader policy and institutional environment in which an intervention is embedded). Outcome is the anticipated and unanticipated consequence that is brought about by the interaction of different interventions' mechanisms in different contexts.

The realist framework will be overlaid with Dervin's sense-making theory, grounded in the user-centred paradigm in information science research, which allows assessment of 'how people make sense of their worlds and how they use information and other resources in the process'.[31] The theory assumes that in the complexity of the real world people constantly face 'discontinuous' situations and breaks in an individual's knowledge exist: this may signal information need and provoke the seeking and use of information to help create new knowledge ('sense').[19 31 32] The sense-making process is operationalised in terms of four elements: the situation, that is, the time–space context in which the information needs arise and sense is constructed; the gap, that is, a cognitive barrier that could be caused by the lack of relevant information or the uncertainty of a situation; the bridge, that is, the strategy supporting the information seeking process that helps to close the gap; the use, that is, the outcome of the information seeking process in terms of how the information obtained or constructed is used and what sense the information enables users to make.[18 19 31]

Dervin's concepts of situation, gap/bridge and use will be adopted to operationalise the realist concept of context, mechanisms and outcomes in the analytical framework.

**Table 2** Overview of the intervention reported in accordance with the Standards for Reporting Implementation Studies checklist

**Intervention (ie, the actual intervention being implemented)**

| Item | Description and reference to methodological frameworks and reporting standards |
|---|---|
| 1. Title | Effectiveness–implementation hybrid type 2 study evaluating an intervention to support 'information work' in dementia care |
| 2. Abstract | See Abstract of the protocol |
| 3. Introduction | See Introduction of the protocol |
| 4. Introduction | See Intervention development of the protocol |
| 5. Aims and objectives | 1. To assess the outcomes of the intervention as perceived by people living with dementia and their family carers, staff implementing the intervention and local organisations and to explain how and why the intervention contributed to achieve these outcomes<br><br>2. To reflect on the extent to which, how and why the co-creation initiative has influenced the design, implementation and impact of the intervention |
| 6. The design and key features of the evaluation | The evaluation of the intervention aims at assessing the outcomes of the intervention with respect to people with dementia and family carers (primary outcomes), to the implementers of the intervention (proximal outcomes) and to local service providers (distal outcomes) |
| 7. Context | As per implementation strategy described above |
| 8. Target population | People living with dementia (any type of dementia, diagnosed at any time) and their family carers attending an appointment with their community psychiatric nurses (CPNs). CPNs were expected to use their clinical judgement in identifying patients or family members deemed to be ready to receive the leaflet |

Continued

**Table 2** Continued

**Intervention (ie, the actual intervention being implemented)**

| Item | Description and reference to methodological frameworks and reporting standards | |
|---|---|---|
| 9. Intervention | Description of the intervention, provided in accordance with the Template for Intervention Description and Replication checklist[46] | Name: Leaflet about local care and support available to people living with dementia and family carers<br>Why: In the diagnostic phase of the co-creation initiative, people living with dementia and family carers consistently reported a lack of timely, up-to-date information provided in a user-friendly format, which hinders their 'information work'. The intervention has been designed to provide a short, up-to-date, paper-based source of information that could be handed out and used by CPNs in their encounter with people with dementia and their family carers and support their 'information work'<br>What (materials): An A4 three-folded, two-sided leaflet printed in high contrast colour combination. Information are organised in four themes, for which relevant services, a short description and contact details are provided<br>What (procedures): CPNs offer the leaflet to patients and their family carers as part of their clinical encounters<br>Who: 27 qualified mental health nurses (CPNs) part of the organic cells working across four locality teams of the local Community Mental Health Services for Older People (CMHSOP) who assess, diagnose, review medications and offer postdiagnostic support to people with dementia and signpost family carers to services available to them<br>How: The leaflet is offered in the context of the appointment attended by the person with dementia and their family carer. The CPN can offer the leaflet in response to a current specific need of the person with dementia or of the family carer (either explicitly reported or latent) or opportunistically, to make them aware of local services which could be of interest either at the present time or in the future<br>Where: The intervention takes place in the outpatient clinics of the CMHSOP services of the local mental health foundation trust<br>When and how much: CPNs can offer the leaflet at any point in the period when the patient is under the care of the CMHSOP (eg, at initial assessment, at diagnosis, in the postdiagnostic appointment, at review, on transfer of care back to primary care (e.g. to general practitioners). It could be offered by the CPNs more than once at different points in time<br>Tailoring: CPNs are expected to use their clinical judgement in deciding whether, when, how and to whom to offer the leaflet based on:<br>▲ patients or families' readiness to accept the leaflet (eg, some patients do not temporarily—or will never—accept a diagnosis of dementia)<br>▲ length of the appointments (eg, initial assessment can take up to 90 min and patients and families may be overloaded with information)<br>▲ type of appointments (eg, titration appointments may cause disappointment in patients) |
| 10. Subgroups | Any subgroups recruited for additional research tasks, and/or nested studies | Not applicable to this study |
| **Evaluation** | | |
| 11. Outcomes | Primary and other outcomes of the intervention | For people with dementia and family carers (primary outcomes)<br>▲ Their satisfaction with the availability of information on local services (eg, in terms of timing and quality of information)<br>▲ Their awareness of local services<br>▲ In the longer term, their access and use of local services<br>For implementers of the intervention (proximal outcomes)<br>▲ CPN, team and service managers' awareness of local services and support available to people diagnosed with dementia and to their family carers<br>▲ CPN, team and service managers' confidence to hold conversations around using services available to those living with dementia and their family carers<br>For local service providers (distal outcomes)<br>▲ Volume of referrals into their services<br>▲ Appropriateness of the use of services among those living with dementia and family carers (eg, with respect to the type and level of need) |

Continued

**Table 2** Continued

Intervention (ie, the actual intervention being implemented)

| Item | Description and reference to methodological frameworks and reporting standards |
|---|---|
| 12. Process evaluation objectives and outcomes related to the mechanism(s) through which the strategy is expected to work | Objectives and outcomes related to the mechanism by which the strategy is expected to work, using the Medical Research Council (MRC) framework for complex interventions[26] |
| 13. Data collection methods | **People with dementia and family carers**<br>▶ Longitudinal semistructured in-depth interviews: interview 1 carried out 4–6 weeks after the appointment with the CPN when they receive the leaflet; interview 2 (follow-up) scheduled about 6 months after interview 1<br><br>**Implementers of the intervention**<br>▶ Semistructured in-depth interviews with CPNs, team managers and service manager of CMHSOP (scheduled about 6 months from the start of the implementation of the intervention)<br><br>**Local service providers**<br>▶ Activity log consisting of a table recording the monthly number of referrals received by the provider in total, by user group (people with dementia or family carer), by referral source (ie, self-referral, CMHSOP, social services and other), by reason of the referral (eg, health and safety assessment, information, support for carer and support for people with dementia), over a 24-month period (ie, in the 12 months preceding the implementation of the intervention and during the 12-month implementation period)<br>▶ Semistructured in-depth interviews with managers (scheduled about 10 months from the start of the implementation of the intervention)<br><br>**Local working group**<br>▶ Focus group with about 12 participants (including commissioners, statutory service providers, staff from CMHSOP, managers from third sector organisations and a family carer) (scheduled about 12 months from the start of the implementation of the intervention) |
| 14. Sample size | **People living with dementia and family carers**<br>▶ Up to 30 people living with dementia (any type of dementia, diagnosed at any time, willing and able to consent) and family carers (of somebody with any type of dementia, any degree of severity and diagnosed at any time)<br><br>**Implementers of the intervention**<br>▶ Up to 12 CPNs from 4 locality teams<br>▶ Four team managers<br>▶ Service manager<br><br>**Local service providers**<br>▶ At least three local organisations out of the seven listed on the leaflet as service providers which in preliminary conversations committed to supply data to inform the evaluation<br><br>**Local working group**<br>▶ About 12 participants attending the focus group (including commissioners, statutory service providers, staff from CMHSOP, managers from third sector organisations, a person with dementia and a family carer) |
| 15. Methods of analysis | Thematic content analysis of interview and focus group data<br>Descriptive statistics for referral data provided by local service providers, triangulated with interview data |

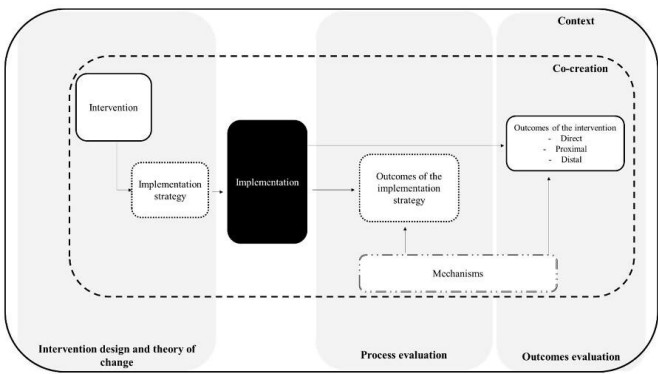

**Figure 3** The evaluation framework of the intervention.

## Data collection methods and samples

The mixed-method evaluation will build on data collected from four sets of stakeholders involved in the intervention (tables 1 and 2, items 13 and 14):

► People living with dementia and family carers who are at the immediate receiving end of the intervention.
► CPNs, team and service managers based in the CMHSOP who are implementing the intervention.
► Local providers of care and support to people with dementia and their families, as the organisations expected to observe a change in their service access and use as a consequence of the intervention.
► Local working group, who contributed to the development of the intervention.

Figure 4 provides an overview of the data collection timeline.

### People with dementia and family carers

Semistructured in-depth interviews with up to 30 people living with dementia (any type of dementia, diagnosed at any time, willing and able to consent) and family carers (of somebody with any type of dementia, any degree of severity, diagnosed at any time) will take place at two points in time: interview 1 will be carried out 4–6 weeks after the appointment with the CPN when they receive the leaflet and interview 2 (follow-up) will be scheduled about 6 months after interview 1.

The two timepoints have been established to best capture participants' experiences of receiving (interview 1) and using (follow-up interview) the leaflet (the interview guides are provided as online supplemental file 1). Accordingly, interview 1 will aim to understand (1)

if, when and how they received the leaflet; (2) whether and how they anticipate using the leaflet; (3) what they think of the leaflet overall (eg, whether it has changed their perceptions and awareness of the local services offering care and support to those living with dementia and their family carers). Interview 2 will aim to understand (1) whether, how, when and why they used (or not) the leaflet; (2) whether and how the leaflet has changed their care seeking behaviour; (3) their satisfaction with the leaflet.

Longitudinal interviews with people living with dementia and their family carers are a well-established research method.[33–37] The follow-up period is appropriate for research purposes. Over a period of 6 months, personal and family circumstances can change and this may prompt those living with dementia and their family carers to seek help and support. Similarly, those recently diagnosed may not feel immediately ready to get in touch with local services but this could change after 6 months.

This purposeful sample will be recruited at the appointment when the CPN hands out the leaflet. The size of the sample is expected to allow to capture a wide range of perspectives (varying for roles, ie, people with dementia vs family carers, types of dementia diagnosis, levels of cognitive impairment, age, gender and living arrangements) while achieving data saturation. The recruitment will be carried out on a rolling basis alongside the implementation of the intervention and will stop when the desired sample size is reached.

### Community Mental Health Services for Older People

In-depth interviews with CPNs, team and service managers from the implementing CMHSOP will be carried out about 6 months from the start of dissemination of the leaflet. Up to 20 staff members will be recruited across the four localities in which the CMHSOP operate.

The interviews aim to understand (1) whether, when and how they used (or not) the leaflet in clinical practice; (2) whether and how the leaflet has changed any aspect of their clinical practice; (3) their views about the impact of adopting the principles of co-creation on the implementation and impact of the intervention. The interview guides are provided as online supplemental file 1.

### Local service providers

Referral data from a minimum of three out of the seven local organisations listed on the leaflet which provide care and support to people living with dementia and their family carers will be collected for a period of 24 months (in the 12 months preceding the implementation and for the 12-month implementation period). Semistructured in-depth interviews with service managers will be undertaken about 10 months from the start of the dissemination of the leaflet to (1) review the referral data and (2) identify and interpret the patterns of referral into each service (eg, over time, with respect to the referral source or the user group) and their appropriateness (eg, with

| Month | | 1 | 2 | 3 | 4 | 5 | 6 | 7 | 8 | 9 | 10 | 11 | 12 |
|---|---|---|---|---|---|---|---|---|---|---|---|---|---|
| Implementation | | | | | | | Ongoing implementation | | | | | | |
| People with dementia and family carers | Recruitment | | | | | | Ongoing recruitment | | | | | | |
| | Data collection | | Interview 1 | | | | | | Interview 2 | | | | |
| Implementers (CPNs, team and service managers) | Data collection | | | | | | Interviews | | | | | | |
| Local service providers | Data collection | | | | | | Activity log | | | | | | |
| | | | | | | | | | | | Interviews | | |
| Local working group | Data collection | | | | | | | | | | | | Focus group |

**Figure 4** Data collection timeline.
CPNs community psychiatric nurses.

respect to the type and level of need). The interview guide is provided as online supplemental file 1.

## Local working group

A focus group with the local working group (about 12 participants including commissioners, statutory service providers, staff from CMHSOP, managers from third sector organisations, a family carer) will be held towards the end of the implementation period, following preliminary analysis of the evaluation data (see online supplemental file 1 for the focus group guide). This will aim to:

► Discuss and validate the emerging findings about the evaluation of the implementation strategy.
► Discuss and validate the emerging findings around the impact of the intervention on people with dementia and their families, CPNs and mental health services and local service providers.
► Reflect on the impact of co-creation initiative on the implementation and impact of the intervention.

## Informed consent

Written and verbal versions of the Participant Information Sheet and Informed Consent Sheet will be presented to participants. The Participant Information Sheet details the nature of the study, what participation will involve, any risks involved. It will be clearly stated that the participant is free to withdraw at any time, for any reason, without impact on future care or service access, and with no obligation to give the reason for withdrawal. Participants will be allowed as much time as they wish to consider the information (at least 48 hours), and will have the opportunity to ask questions to the investigator, their General Practitioner (GP) and/or CPNs (for people with dementia and family carers), or other independent parties, in order to decide whether they will participate in the study. Written informed consent will be obtained.

Maximum efforts will be devoted to recruit people living with dementia into this study. Historically, people with dementia have been excluded from much research in social and healthcare on the basis that their declining cognitive abilities may affect capacity to consent.[38] However, the ability of many people with dementia to provide informed consent, based on their ability to understand information, evaluate risks and benefits, and communicate decisions effectively, has been established.[39 40] Prior to the interview, an informal conversation about the research will allow the researchers to ascertain that the potential participant has capacity to consent. If the researcher believes that the person with dementia has capacity, she will take written informed consent and will start the interview.

## Data analysis

The referral data provided by each local provider will be collated, tabulated and described with simple metrics (eg, number of referrals in total, by user group, by referral source, by reason for referral). The results will be triangulated with interview data collected from the participants

living with dementia and family carers, the implementers and the managers of the local service providers.

The interviews will be transcribed, anonymised and analysed using thematic analysis (performed with NVivo12) (tables 1 and 2, item 15).[41] Following the realist approach, the initial codebook will be developed around the themes (codes) of context conditions, underlying mechanisms, observed outcomes matched on to implementation strategy and intervention. These will be then refined into subcodes to capture specificities of data collected from different interviewee roles or to highlight different dimensions featuring in a main code. For example, for intervention strategy, the subcodes of 'fidelity', 'dose', 'adaptations', 'reach' may be developed or 'mechanisms' may be clustered around the subcode 'ideas'.

## Patient and public involvement

A person living with dementia and a family carer of somebody living with dementia are members of the advisory board of the co-creation initiative. In this capacity they are overseeing the development of the study, provided feedback in the design phase of the intervention, reviewed the interview guides and are expected to review the results.

## ETHICS AND DISSEMINATION

Ethical and research governance approvals have been obtained from the West Midlands—South Birmingham Research Ethics Committee (REC reference 16/WM/0397).

The study has been designed as part of a 5-year co-creation initiative by the local working group in collaboration with the research team. Updates about the implementation of the intervention are regularly shared with the working group and with the wider group of stakeholders involved in the co-creation initiative.

Findings of the work will be disseminated across professionals, for example, at national conferences and events for providers and commissioners of care and support for people with dementia, policy-makers in the area of dementia, representatives of services users. Articles stemming from this work will be submitted for publication to peer-reviewed journals and to national and international scientific conferences in the field of applied health research.

**Contributors** CDP was involved in the design of the intervention and of the implementation strategy, conceived the evaluation framework, drafted the first version of the protocol and revised it following discussions with the other authors. JRO and MA oversaw the design of the intervention and of the implementation strategy, contributed to developing the evaluation framework and revised the manuscript. CB was involved in the design of the intervention and of the implementation strategy, supported local engagement and revised the manuscript. RG was involved in the design of the intervention and of the implementation strategy, supported local engagement and revised the manuscript. All authors have read and approved the final manuscript.

**Funding** CDP, JRO, RG and MA were supported by The Health Foundation, grant number 1274233.

**Competing interests** None declared.

**Patient and public involvement** Patients and/or the public were involved in the design, or conduct, or reporting, or dissemination plans of this research. Refer to the Methods and analysis section for further details.

**Patient consent for publication** Not required.

**Provenance and peer review** Not commissioned; externally peer reviewed.

**Open access** This is an open access article distributed in accordance with the Creative Commons Attribution 4.0 Unported (CC BY 4.0) license, which permits others to copy, redistribute, remix, transform and build upon this work for any purpose, provided the original work is properly cited, a link to the licence is given, and indication of whether changes were made. See: https://creativecommons.org/licenses/by/4.0/.

**ORCID iD**
Chiara De Poli http://orcid.org/0000-0002-1879-553X

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
