## [Reviewer comments · BMJ Open]

ARTICLE DETAILS

TITLE (PROVISIONAL)	Effectiveness-implementation hybrid type 2 study evaluating an intervention to support 'information work' in dementia care: implementation study protocol
AUTHORS	De Poli, Chiara; Oyebode, Jan; Binns, Christopher; Glover, Richard; Airoldi, Mara

VERSION 1 – REVIEW

REVIEWER	Nicole Dalmer Department of Health, Aging and Society Faculty of Social Sciences McMaster University
REVIEW RETURNED	02-Apr-2020

GENERAL COMMENTS	My thanks to the authors for submitting this implementation study protocol – the information needs and practices of family caregivers and those living with dementia is an understudied area that is in desperate need of additional research. I am particularly encouraged with the co-creation approach to ensure that the information pamphlets can be of maximum value to the stakeholders involved. On this co-creation note, given the paper's emphasis on the collaborative development of the intervention, I am curious why family caregivers and individuals living with dementia were cast in the role of 'consultants' and were not part of the initial local working group? I'm wondering if the intervention might have appeared differently if family carers and individuals living with dementia were part of initial the prototype development instead of testing the already-developed three prototypes? Because the focus is very much on information work and on family caregivers and older adults' behaviours with information (information pamphlets, in particular), one of my chief pieces of feedback surrounds the need to integrate literature from the discipline of information studies. In particular, there are several readings that get at the particulars of information work – that is, a more critical lens in considering the many lines of work (such as emotional labour) of using and dealing with the implications of information throughout the dementia progression as well as the importance of considering temporality with regards to individuals' information behaviours. In particular, I was surprised to not find the following key readings referenced in the paper and wonder if these might be of use as the study unfolds and in helping to nuance the analysis of the pamphlet's effectiveness and utility: • Harland, J. A., & Bath, P. A. (2008). Understanding the information behaviours of carers of people with dementia: A critical review of models from information science. Aging and Mental Health, 12(4), 467-477.
---

	 • Pálsdóttir, Á. (2017). From noticing to suspecting: The initial stages in the information behaviour of informal caregivers of people with dementia. In International Conference on Human Aspects of IT for the Aged Population (pp. 452-466). Springer, Cham. • Barnes, M., Henwood, F., Smith, N., & Waller, D. (2013). External Evaluation of the Alzheimer's Society Carer Information and Support Programme (CrISP). University of Brighton, Brighton, UK. • Harris, R. M., & Dewdney, P. (1994). Barriers to information: How formal help systems fail battered women. Westport, Conn.: Greenwood Press. (This is a seminal publication that might be helpful in understanding how individuals access and process information from formal health-related systems). Perhaps my chief concern and curiosity lies in the authors' decision to combine family caregivers and individuals living with dementia – these two groups consistently appear together throughout the document. From my understanding, one informational pamphlet has been designed for both family caregiver and individual living with dementia – which has the potential to conflate each group's vastly different information needs and practices (and different lines of work within the information work umbrella). I'm wondering if the pamphlet is being translated or modified for each of these two populations? I am very much looking forward to keeping abreast with the study's evolving findings and conclusions.
--	--

REVIEWER	Maria Pertl Royal College of Surgeons in Ireland
REVIEW RETURNED	05-Jun-2020

GENERAL COMMENTS	This is a well and clearly written protocol, that presents the project in considerable detail following best practice guidelines and reporting standards. Some minor comments and considerations are detailed below. In Table 2, section 5 on the Aims and objectives is the same for both the Implementation Strategy and the Intervention. It would be clearer to state the specific Aims and objectives that fall under each of these two aspects of the study. Also in Table 2, page 12, 'team' should be plural under 'Who'. On page 18, under Local service providers, it would be helpful to provide more detail on what and how activity data will be collected within the activity log – will there be a template to standardise this across organisations? Is the main focus here just on the volume of referrals? Also, how will data on “a change in pattern of referral to services” be collected? Does this include the “appropriateness of the use of services” alluded to in Table 2? How will this be determined? What is the justification of a sample size of “up to five organisations”? Is there a minimal number of organisations that will be deemed sufficient and how will the service providers be selected? How will improvements in awareness of local services be established among people living with dementia and family carer and, especially, implementers of the intervention since these are only being interviewed once? In table 2, the timing of the
--

	interviews with implementers or the local working group is not specified under Intervention, section 13, as it is for the other participants. More detail on the nature of the interview guides, with some sample questions, would be helpful. Will there be PPI input into the construction of the interview guides? What steps will be taken to validate the data? The Data analysis section on page 19, makes no mention of how the activity data from local service organisations will be analysed or how triangulation will be carried out.
--	---

VERSION 1 – AUTHOR RESPONSE

Reviewer: Nicole Dalmer, Department of Health, Aging and Society, Faculty of Social Sciences, McMaster University, Canada

Dear Dr Dalmer,

Many thanks for reviewing our article and for your thoughtful comments. We provide below our response to the points that you raised. We have addressed each of them in the manuscript as appropriate.

- 1. I am particularly encouraged with the co-creation approach to ensure that the information pamphlets can be of maximum value to the stakeholders involved. On this co-creation note, given the paper's emphasis on the collaborative development of the intervention, I am curious why family caregivers and individuals living with dementia were cast in the role of 'consultants' and were not part of the initial local working group? I'm wondering if the intervention might have appeared differently if family carers and individuals living with dementia were part of the initial prototype development instead of testing the already-developed three prototypes?**

Thanks for this comment. We have been very committed to involve people living with dementia and family carers throughout the co-creation initiative, however their recruitment and engagement with the study activities have proved to be challenging (as we documented in a related journal article currently being published)[1].

We reviewed our fieldnotes and were reminded that a local third sector organization identified two family carers keen to join the working group developing the leaflet. Both were invited. One could not attend the meetings of the working group due to competing commitments but the other was fully involved. To mitigate the limited involvement of people with dementia and family carers, the working group involved front line staff, including one person who was also a family carer, from local third sector organizations providing daily direct support and care to people with dementia and to family carers. In a similar vein, the professional designer also had extensive experience and expertise in producing written information material for people with cognitive impairment.

We shared our concerns about the lack of user's representation with the working group and, subsequently, we agreed to involve those living with dementia and their family carers via a consultation exercise. To maximize the opportunities to reach them, the consultation exercise was organized around the times and places when activities or groups for people with dementia and/or family carers members were scheduled to take place. We felt that in this way we could improve their participation in the process.

We have revised some paragraphs of the section on the Intervention and the implementation strategy to provide more information about the membership of the working group and about the consultation process.

- 2. Because the focus is very much on information work and on family caregivers and older adults' behaviours with information (information pamphlets, in particular), one of my chief pieces of feedback surrounds the need to integrate literature from the discipline of information studies. In particular, there are several readings that get at the particulars of information work – that is, a more critical lens in considering the many lines of work (such as emotional labour) of using and dealing with the implications of information throughout the dementia progression as well as the importance of considering temporality with regards to individuals' information behaviours. In particular, I was surprised to not find the following key readings referenced in the paper and wonder if these might be of use as the study unfolds and in helping to nuance the analysis of the pamphlet's effectiveness and utility:**

Harland, J. A., & Bath, P. A. (2008). Understanding the information behaviours of carers of people with dementia: A critical review of models from information science. *Aging and Mental Health*, 12(4), 467-477.

Pálsdóttir, Á. (2017). From noticing to suspecting: The initial stages in the information behaviour of informal caregivers of people with dementia. In *International Conference on Human Aspects of IT for the Aged Population* (pp. 452-466). Springer, Cham.

Barnes, M., Henwood, F., Smith, N., & Waller, D. (2013). External Evaluation of the Alzheimer's Society Carer Information and Support Programme (CrISP). University of Brighton, Brighton, UK.

Harris, R. M., & Dewdney, P. (1994). *Barriers to information: How formal help systems fail battered women*. Westport, Conn.: Greenwood Press. (This is a seminal publication that might be helpful in understanding how individuals access and process information from formal health-related systems).

Thanks for providing this background literature. Given the word count limit of the article, we are not able to provide a comprehensive overview of the literature on information behaviour and information work to position our work. However, following your suggestion, we have revised the Introduction to acknowledge that information work has been addressed both in empirical studies and in theoretical work. We have also revised the analytical framework to embed Wilson's model of information need, seeking and use. The key dimensions of this model will be used to operationalize our analytical framework.

- 3. Perhaps my chief concern and curiosity lies in the authors' decision to combine family caregivers and individuals living with dementia – these two groups consistently appear together throughout the document. From my understanding, one informational pamphlet has been designed for both family caregiver and individual living with dementia – which has the potential to conflate each group's vastly different information needs and practices (and different lines of work within the information work umbrella). I'm wondering if the pamphlet is being translated or modified for each of these two populations?**

Thanks for raising this point, which had been matter of extensive discussion among the leaflet working group. It was put forward as a question also during the consultation exercise with people with dementia and family carers and was discussed with the advisory board of the project. In the current phase of the study we intend to pilot one version only of the leaflet that aims to provide information to both people with dementia and family carers. The rationale for this choice is manifold:

- Local stakeholders and people with dementia and family carers who took part in the consultation converged on the idea of producing a leaflet with essential information about and contact details of the key local organizations that provide dementia-related support and services. The leaflet is intended mainly to signpost those living with dementia and family carers to local organizations and, in fact, offers an overview of these organizations,

not a detailed list of their services. In practice, most organizations provide services for both groups. Hence two versions of the leaflet would have had very similar content.

- Two versions of the leaflet would not be any different in terms of graphic design either, as we assumed similar needs for those living with dementia and family carers in terms of readability and accessibility of the text and ease of navigation of the written material.
- In response to issues of information overload experienced by those living with dementia and their families, we wanted to avoid giving too much detail; hence one single document with concise information relevant to both groups was deemed to suffice.
- From the implementation point of view, staff in a position to embed the leaflet in their clinical practice felt that it would be simpler to hand out only one version of the leaflet, even more so as the person with dementia was very often accompanied by a family member.
- The working group acknowledged that designing and using one version of the leaflet would, more likely, contribute to its sustainability in the longer term. The updating and version-controlling processes were expected to be less burdensome if one single version was available.

Following your comment, we have added a paragraph in the section The intervention and the implementation strategy to explain the rationale for this choice. We take this opportunity to emphasise that, depending on the results of the evaluation, the working group will consider ideas to improve the leaflet in the future, for example, by producing different versions for different stakeholder groups (i.e. one for people living with dementia and one for family carers, as suggested) or for more granular geographical areas (e.g. at the level of locality instead of at the level of a macro-area).

References

- 1 De Poli C, Oyebode JR, Airoidi M, *et al.* A need-based, multi-level, cross-sectoral framework to explain variations in satisfaction of care needs among people living with dementia. *BMC Health Serv Res* (forthcoming)

Reviewer: Maria Pertl, Royal College of Surgeons in Ireland, Ireland

Dear Dr Pertl,

Many thanks for reviewing our article and for your thorough comments. We provide below our response to the points you raised. We have addressed them in the manuscript as appropriate.

1. **In Table 2, section 5 on the Aims and objectives is the same for both the Implementation Strategy and the Intervention. It would be clearer to state the specific Aims and objectives that fall under each of these two aspects of the study. Also in Table 2, page 12, ‘team’ should be plural under ‘Who’.**

Thanks for this comment. We revised item “Aims and objectives” in Table 1 and Table 2. We also rectified the typo.

2. **On page 18, under Local service providers, it would be helpful to provide more detail on what and how activity data will be collected within the activity log – will there be a template**

to standardise this across organisations? Is the main focus here just on the volume of referrals? Also, how will data on “a change in pattern of referral to services” be collected? Does this include the “appropriateness of the use of services” alluded to in Table 2? How will this be determined?

Thanks for these questions. The activity log consists of a table recording on a monthly basis the number of referrals received by the provider in total, by user group (people with dementia or family carer), by referral source (i.e. self-referral, CMHSOP, social services, other), by reason for referral (e.g. health and safety, information, support for carer, support for people with dementia). These data will be reviewed with the service managers from each provider organization to identify and interpret the pattern of referral in to each service (e.g. over time, with respect to the referral source or the user group) and their appropriateness (e.g. with respect to type and level of need).

We have added more details about these aspects in the manuscript and in the tables.

3. What is the justification of a sample size of “up to five organisations”? Is there a minimal number of organisations that will be deemed sufficient and how will the service providers be selected?

Thanks for these questions. We have reviewed this decision and have stated our aimed-for sample size as ‘at least three of the seven local provider organisations listed on the Leaflet. As this approaches 50%, this would give useful data to gain a picture of change. After preliminary conversations with the local providers listed on the leaflet, five out of seven have already committed to supply activity data to inform the evaluation.

4. How will improvements in awareness of local services be established among people living with dementia and family carer and, especially, implementers of the intervention since these are only being interviewed once?

Thanks for these questions. We will be unable to measure improvements in awareness of services within individual participants. However, we do have a general baseline gauge of awareness from the diagnostic phase of the study. This highlighted that the vast majority of people with dementia and family carers interviewed lacked a reliable source of information about local organizations providing care and support services, some relied on word-of-mouth or serendipitous advice and others missed out. The leaflet intends to fill this gap. Hence the evaluation intends to explore whether the leaflet has facilitated information and care seeking behaviour of people with dementia and families and their awareness of the local services available to them. The interview content will capture awareness of services as a distinct outcome from actual use of services, as each reflects different stages in the care seeking process,[1] and their longitudinal design aims to check whether any changes in awareness and service use occur over time (the follow up period is dictated by the duration of the funding of the study). The interview guides have been designed accordingly.

Conversations with the working group (and other local stakeholders involved in the co-creation initiative) highlighted that usually professionals have a broad knowledge of how the local dementia care system is organized (e.g. what the key organizations are, their remits and areas of specialization), which is assumed as a baseline of the evaluation. The interviews planned with the implementers aims to explore whether and the extent to which the leaflet has shifted this.

We have added more details about these aspects in the manuscript and in the tables.

5. In table 2, the timing of the interviews with implementers or the local working group is not specified under Intervention, section 13, as it is for the other participants.

Thanks for spotting this inconsistency. We revised the table to include the timepoints when the data collection will take place. We revised also Figure 4 to ensure its consistency with item 13 in Table 2.

6. More detail on the nature of the interview guides, with some sample questions, would be helpful. Will there be PPI input into the construction of the interview guides? What steps will be taken to validate the data?

The interview guides were reviewed by the advisory board of the project, which includes a family carer, a person living with dementia, local commissioners and service providers. The interview guides are provided as supplementary material. A focus group with the local working group will be used to validate the data generated with the interviews, carried out with the different group of stakeholders, as outlined in the sub-section Local working group.

7. The Data analysis section on page 19, makes no mention of how the activity data from local service organisations will be analysed or how triangulation will be carried out.

Thanks for this comment. The referral data provided by each local organization will be collated, tabulated and described with simple metrics (e.g. number of referrals in total, by user group, by referral source, by reason for referral). We will triangulate the results of the analysis of the referral data with the interview data.

We have added more details about these aspects in the manuscript and in the tables.

References

1 De Poli C, Oyebode JR, Airoldi M, *et al.* A need-based, multi-level, cross-sectoral framework to explain variations in satisfaction of care needs among people living with dementia. *BMC Health Serv Res*

VERSION 2 – REVIEW

REVIEWER	Nicole Dalmer McMaster University Canada
REVIEW RETURNED	Canada 30-Jul-2020

GENERAL COMMENTS	Thank you for the opportunity to review this manuscript again. The authors have carefully considered and responded to feedback, and the manuscript is much stronger. I have one small amendment to suggest and one larger issue to take into consideration: A small amendment: One sentence in the introductory paragraph ("Also, it assumes patients and family carers are physically and cognitively able to carry out information work, and it overlooks the emotional processes that influence adaptation to long-term conditions, which may influence whether information work is successfully undertaken") is paraphrased from Dalmer's 2018 work ("Informing care: Mapping the social organization of families' information work in an aging in place climate") and should be attributed to that work.
--

	As for the larger issue, one remaining element that may warrant some rethinking is the analytical framework, particularly with regards to the application and integration of models and frameworks from library and information sciences. Wilson's information seeking behaviour model is not generally called as such (especially among his peers and colleagues in the library and information science community) – we generally refer to his model as Wilson's model on information behaviour. Furthermore, as information behaviours are generally more closed and are considered less contextualized than information practices, I would urge the authors to instead consider drawing on and integrating Pam McKenzie's landmark writing (2003) on information practices ("A model of information practices in accounts of everyday-life information seeking"). As I understand from the authors' writing, an information practices lens is more in keeping with the participants' experiences of information, as this lens recognizes that information activities are socially structured and mediated practices and it takes into account the institutional structures that shape, foster, or inhibit information practices. Alternatively, if the authors are focused on information seeking and the many micro and macro influences on our everyday information seeking (which seems to be the case), might they instead consider Brenda Dervin's sense-making theory. This is another key, foundational information seeking/information practices model in library and information science and appears to also align with the intent and findings in this article.
--	---

VERSION 2 – AUTHOR RESPONSE

Reviewer: Nicole Dalmer, Department of Health, Aging and Society, Faculty of Social Sciences, McMaster University, Canada

Dear Dr Dalmer,

Many thanks for reviewing our article and for your comments. Below we provide our response to the points that you raised and explain how we have addressed them in the manuscript.

- 1. One sentence in the introductory paragraph ("Also, it assumes patients and family carers are physically and cognitively able to carry out information work, and it overlooks the emotional processes that influence adaptation to long-term conditions, which may influence whether information work is successfully undertaken") is paraphrased from Dalmer's 2018 work ("Informing care: Mapping the social organization of families' information work in an aging in place climate") and should be attributed to that work.**

Thanks for bringing to our attention this work, we were not aware of it. We have added the relative reference, as suggested.

- 2. As for the larger issue, one remaining element that may warrant some rethinking is the analytical framework, particularly with regards to the application and integration of models and frameworks from library and information sciences. Wilson's information seeking behaviour model is not generally called as such (especially among his peers and**

colleagues in the library and information science community) – we generally refer to his model as Wilson’s model on information behaviour. Furthermore, as information behaviours are generally more closed and are considered less contextualized than information practices, I would urge the authors to instead consider drawing on and integrating Pam McKenzie’s landmark writing (2003) on information practices (“A model of information practices in accounts of everyday-life information seeking”). As I understand from the authors’ writing, an information practices lens is more in keeping with the participants’ experiences of information, as this lens recognizes that information activities are socially structured and mediated practices and it takes into account the institutional structures that shape, foster, or inhibit information practices. Alternatively, if the authors are focused on information seeking and the many micro and macro influences on our everyday information seeking (which seems to be the case), might they instead consider Brenda Dervin’s sense-making theory. This is another key, foundational information seeking/information practices model in library and information science and appears to also align with the intent and findings in this article.

Thanks for providing further suggestions about possible theories and models of information behaviour to help strengthen our work. We have now revised the analytical framework to embed Dervin’s sense-making theory of information need, seeking and use. We intend to adopt the key dimensions of this theory, in the configuration of situation-gap/bridge-use, to operationalize our analytical framework.